# Magnitude of syphilis sero-status and associated factors among pregnant women attending antenatal care in Jinka town public health facilities, Southern Ethiopia, 2020

**Mulusew Enbiale[1], Asmare Getie[2]\*, Frehiwot Haile[1], Beemnet Tekabe[1], Diresign Misekir[1]**

1 School of Public Health, Arba Minch University, Arba Minch, Ethiopia, 2 School of Nursing, Arba Minch University, Arba Minch, Ethiopia

\* asmaregetie2017@gmail.com

**Data Availability Statement:** All relevant data are within the manuscript and its Supporting Information files.

## Abstract

### Introduction

Syphilis is one of the leading causes of perinatal morbidity and mortality and is one of the most important public health problems. There was no study showing syphilis serostatus and its related factors among pregnant women in the current study area. This study was aimed to assess the magnitude of syphilis serostatus and associated factors among pregnant women attending antenatal care in Jinka town public health facilities.

### Method

Institution based cross-sectional study design was conducted in Jinka town public health facilities, southern Ethiopia from the 1st July to the 1st September, 2020. A systematic sampling technique was used to select 629 study subjects. Data were collected using a structured questionnaire through face-to-face interviews and records were reviewed to check syphilis test results. Data were coded and entered by using Epi-data version 4.432 and analyzed using SPSS version 25. The binary logistic regression model was used to investigate factors associated with syphilis. A p-value of < 0.05 at multivariable analysis was considered statistically significant.

### Result

In this study, syphilis sero-prevalence among pregnant women attending antenatal care clinics was 4.8% (95% CI: 3.12, 6.48). Rural residence [AOR: 2.873; 95%CI (1.171, 7.050)], alcohol use [AOR: 3.340; 95% CI (1.354, 8.241)] and having multiple sexual partner [AOR: 5.012; 95% CI (1.929, 13.020)] were statistically significantly associated with syphilis.

### Conclusion

Sero-prevalence of syphilis was high. Being a rural residence, having multiple sexual partners, alcohol use were factors associated with syphilis. Therefore, substantial efforts have

**Funding:** The author(s) received no specific funding for this work.

**Competing interests:** The authors have declared that no competing interests exist.

**Abbreviations:** ANC, Antenatal Care; OR, Adjusted odd ratio; CS, Caesarean section; COR, crude odd ratio; IRB, Institutional review board; MTCT, Prevention of Mother to Child Transmission; NR, Non-reactive; RPR, Rapid Plasma Reagan; SNNPR, South Nation Nationalities People Region; SSA, Sub-Saharan Africa; STI, Sexually Transmitted Infection; SDG, Sustainable Development Goal; VDRL, Venereal Disease Research Laboratory.

to be made to provide regular health education for pregnant women at the antenatal clinic on the avoidance of risky behaviors and the risk of syphilis on their pregnancy.

## Introduction

Syphilis is a chronic systemic infection caused by the spirochete Treponema pallidum, which can be transmitted sexually and during a blood transfusion (acquired syphilis), and vertically (congenital syphilis) through the mother's placenta to the fetus [1]. Pregnant women who are infected with syphilis can transmit the infection to their fetus, causing congenital syphilis. The majority of pregnant women with syphilis are not detected and treated early enough to avoid the adverse effects of infection on their pregnancy. There is also antenatal care coverage and syphilis testing variation. It remains an important global public health problem, and its incidence is increasing in different parts of the world [2–4].

Early diagnosis and treatment of syphilis in pregnancy are well-recognized as an effective strategy to reduce syphilis transmission and adverse pregnancy outcomes due to untreated maternal syphilis. In endemic countries, antenatal screening for syphilis detection and treatment can reduce the number of stillbirths by 82%, preterm birth by 64%, and neonatal deaths by 80% [5].

Several studies in different parts of the world have shown that syphilis in pregnancy continuing as a public health concern. A report on global sexually transmitted infection surveillance indicated that the sero-prevalence of syphilis among ANC attendees was 0.8% globally. According to this report, in WHO regions of the Americas and African regions showed the sero-prevalence of syphilis was 0.7% and 2% respectively [6]. A study done in Tanzania revealed that 2.5% of pregnant women attending ANC were seropositive for syphilis [7]. In Ethiopia, the 2014 ANC based sentinel surveillance report showed that sero-prevalence of syphilis was 1.2% [8].

Syphilis is a major preventable and treatable contributor to infant morbidity and mortality. Worldwide, about 53% to 82% of untreated maternal syphilis cases result in adverse pregnancy outcomes including early fetal loss, stillbirth, preterm birth, low birth weight, and neonatal death [9]. It results in a global congenital syphilis rate of 473 per 100,000 live births and the African region an estimated CS case rate of 1,119 per 100,000 live births which accounted for 62% of the total CS cases [10].

Maternal age, residence, educational level, educational level of husband, occupational status, occupational status of the husband, number of pregnancy, history of abortion, history of STI, and HIV/AIDS status have been some of the factors studied and reported in different epidemiologic studies as they affect the magnitude of syphilis serostatus. However, there are important contributing factors like alcohol use, knowledge about STI, having two or more sexual partners which were given little attention and thus are not well studied and understood [11, 12].

Globally, the coverage of ANC first visit, syphilis screening, and treatment among pregnant women in 2016 were, 88%, 66%, and 78% respectively. In the African region, the coverage of ANC first, syphilis screening, and treatment were, 83%, 47%, and 76% consequently [10]. In Ethiopia, ANC first and syphilis screening coverage was 62% and 44.6% respectively [13, 14].

Despite these efforts, syphilis is still one of the leading causes of perinatal morbidity and mortality in most of the developing countries including Ethiopia and is one of the most important public health problems. In Ethiopia, the 2012 and 2014 ANC based sentinel surveillance

report showed a slight increment of syphilis serostatus from 1.0% in 2012 to 1.2% in 2014 [8]. Therefore, a proper understanding of associated factors may give evidence to plan an intervention on these determinants, improve treatment compliance, and improve health promotion strategies in a variety of contexts. Up-to-date evidence for determinant factors associated with syphilis among pregnant women in Ethiopia as well as in the study area is essential.

To the best of our knowledge, the availability of the study on syphilis and its associated factors is limited in Southern Ethiopia. And at present, there is no study on the burden and associated factors of syphilis among pregnant women in Jinka town. This study was aimed to fill this gap by assessing the seroprevalence of syphilis in pregnant women and identifying associated factors among the target women in Jinka town public health facilities, Southern Ethiopia.

## Methods

### Study area and priod

The study was conducted in Jinka town public health facilities from the 1$^{st}$ July to the 1$^{st}$ September 2020. Jinka is the capital city of the south Omo zone, southern Ethiopia. It is located 733km from the capital city of Ethiopia (Addis Ababa) and 462km from Hawassa which is the capital city of SNNPR. It has a latitude and longitude of 5˚47'N 36˚34'E /5.783˚N 36.567˚E and an elevation of 1490 meters above sea level. According to the town health office report, the total population projected for 2020 is 42,219, of whom 21,531 are females. The number of females in the reproductive age group (15–49 years) in the town were 10,061. The government health facilities in this town were one hospital, one health center, six health posts.

### Study design

An institution-based cross-sectional study was used.

### Population

**Source population.**    All pregnant women attending ANC in public health facilities of Jinka town.

**Study population.**    Pregnant women attending ANC clinic in public health facilities which fulfill the inclusion criteria during the study period.

### Inclusion and exclusion criteria

**Inclusion criteria.**    Pregnant women attending antenatal care visits were eligible for the study.

**Exclusion criteria.**    Pregnant women who were seriously ill at the time of the study period were excluded.

### Sample size determination

The sample size for the sero-prevalence of syphilis was determined using a single population proportion formula, assuming a 95% confidence level and by taking 5.1% sero-prevalence of syphilis from a previous study conducted at Yirgalem, Southern Ethiopia [15] and considering 10% non-response rate. The final sample size was 512.

The sample size determination using factors associated with syphilis serostatus among pregnant women attending antenatal care, was calculated by Epi Info 7 Stat Calc program, 2020 using the assumptions and it was 629. The sample size calculated using predictor variables was greater than the sample size determined using the prevalence of syphilis from a previous study. So, by taking the larger number, the final sample size for the study was 629.

## Sampling technique and procedure

The public health facilities in the study area were one hospital and one health center. All ANC visit pregnant women who came to these two health facilities were included in the study. The proportionate allocation method was used to assign the number of pregnant women to each health facility based on the flow of pregnant women per month taking the last sixth months HMIS antenatal records. Finally, a systematic random sampling technique was applied to select 629 study participants.

The number of pregnant women who have attended antenatal care in the study public health facilities in the last two months were 1272. The sample was proportionally allocated to the two public health institutions (**Fig 1**).

## Data collection tool and procedure

A face-to-face interview was conducted using a semi-structured, pretested and standardized questionnaire to obtain data about socio-demographic, obstetric, medical, and behavioral conditions. The questionnaire has included questions that contain all the required data. The questionnaire was initially developed in English and then translated to the Amharic language for ease of communication with the study participants, and translated back to English to confirm consistency. Data were collected after obtaining informed consent from the study participants. Records were reviewed to check syphilis and HIV test results. Data were collected by three female clinical and one male BSc midwives and supervised by one female BSc midwife and one male public health officer selected from facilities. The training was given on the data collection process for a day.

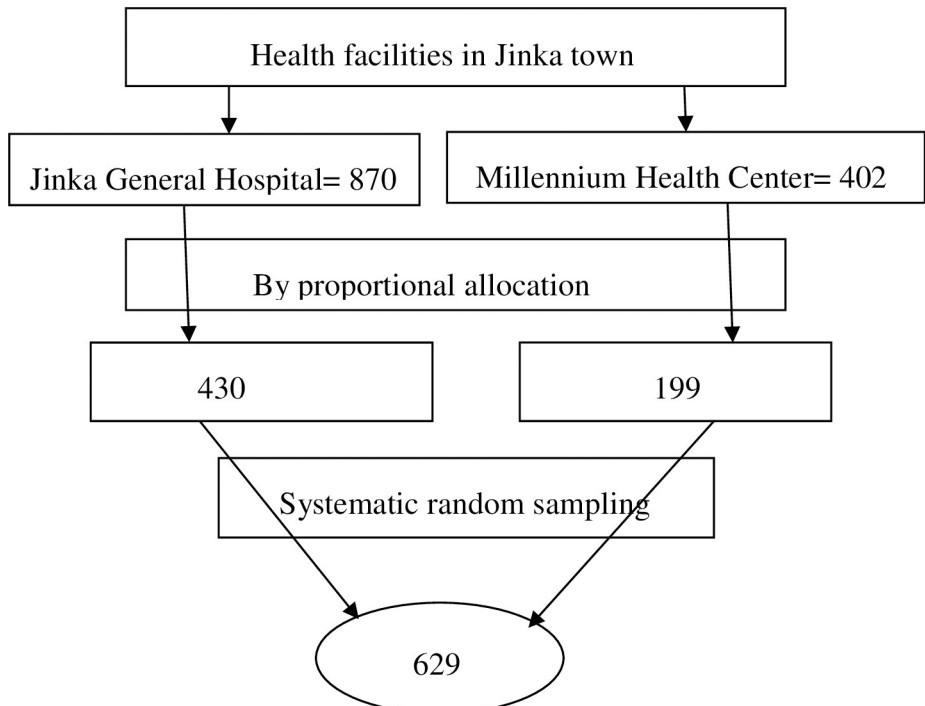

**Fig 1. Schematic diagram for how the pregnant women were selected for a study of the magnitude of syphilis and associated factors among pregnant women in public health facilities of Jinka town, southern Ethiopia, 2020.**

## Study variables

**Dependent variable.** Syphilis serostatus.

**Independent variables.** *Socio-demographic variable*. Age, residence, educational level, educational level of husband, occupational status, occupational status of the husband.

*Obstetric related variables*. Gravidity, history of abortion.

*Medical-related variables*. Previous history of STI, HIV/AIDS status.

*Behavior related variables*. Alcohol use, multiple sexual partners.

**Operational definitions.** *Case Definition of Syphilis*. In this study, a case of syphilis was considered when Rapid Plasma Reagin (RPR) or Venereal Disease Research Laboratory (VDRL) test was reactive for syphilis infection among pregnant women [16].

*Knowledge of STI*. six questions were used for assessing the level of knowledge. The total score of each study participant was converted to a percentage and used to categorize them into those with good knowledge (score> = 50) and poor knowledge (score <50) [17].

*Alcohol use*. consuming alcohol at least once per month in the last twelve months (above the recommended level) [13].

*Multiple sexual partners*. having two or more sexual partner in the lifetime

## Data quality management

To ensure data quality, one-day training was given for the data collectors and supervisors to create a common understanding. Before the actual data collection, a pre-test was conducted in Koybe hospital on 32 individuals (5%) using a structured questionnaire. Based on the finding necessary correction was made. The principal investigator and supervisors were actively involved in the supervision of the data collection. Data collectors were instructed to check the completeness of each questionnaire whether each question was completely answered and the supervisors rechecked the completeness of the questionnaire immediately after submission. The filled questionnaires were checked daily for completeness by the supervisors and principal investigator.

## Data processing and analysis

Data were coded and entered into EPI data version 4.432 and then exported to SPSS version 25 for analysis. Data cleaning was performed by running the frequency of each variable to check the accuracy, inconsistency, and missed value of the data. Descriptive statistics were done and summarized using texts, tables, and graphs based on the type of variables.

In bivariable logistic regression analysis variables having P-value $\leq$ 0.2 was a potential candidate for multivariable logistic regression analysis to control confounders. The degree of association between independent and dependent variables was assessed by using an odds ratio with a 95% confidence interval and variables having P-value <0.05 in the multivariable logistic regression model were considered as statistically significant. Model fitness was checked by Hosmer and Lemeshow Goodness of fit test (p-value = 0.936) and multi-collinearity was assessed using the method variance inflation factors.

## Ethics approval and consent to participate

The study protocol was approved by the Institutional Review Board (IRB) of the College of Health Sciences, Arba Minch University with meeting number 203/2020. A letter of permission was obtained from the South Omo zone health office and Jinka town administration health office and letter of cooperation were also written to the study health facilities. Data collectors approached the study participants by keeping physical distance, using a face mask, and

sanitizer to protect COVID-19. ANC clients were informed about the purpose of the study and written informed consent was obtained before data collection. The information collected is kept confidential.

## Result

### Socio-demographic characteristics of participants

A total of 629 pregnant women attending antenatal care in public health facilities were invited to participate in the study. Of these, 624 pregnant women participated in the study making the response rate of 99.2%. The age range of the study participants was from 18–42 years and the mean (±SD) age of respondents was 25.68 (±4.402) years. The majority of 588(94.2%) of study participants were married and 306(49%) of the participants attended secondary education and above. More than three fourth (79.3%) of the pregnant women were urban residents and 246 (39.6%) were housewives. More than one third (36.1%) of participants earned >3,000 ETB (Table 1).

### Obstetric characteristics of participants

From the total study participants four hundred twenty (67.3%) of the study participants were multigravida. Out of 624 respondents, the majority 479 (76.8%) were screened for syphilis in the second trimester, whereas 94 (15.1%) and 51 (8.2%) of them were screened for syphilis in the first and third trimesters of their pregnancy respectively. Nearly one-third (34.8%) of the participants had ANC first visit, while ninety-nine (15.9%) of them were in their fourth ANC visit. From the total respondents fifty-nine (9.5%) of participants reported a history of abortion.

### Medical characteristics of participants

From the total respondents sixty-one (9.8%) of the participants were reactive for HIV serostatus. Among these, thirteen (21.3%) women knew their HIV status for the first time (**Table 2**).

### Behavior related characteristics

Of the total respondent's two hundred seventy-three (43.8%) of the participants had two or more sexual partners in their lifetime (**Table 3**).

### Knowledge related characteristics

Among 624 study participants, 275(44.1%), 235(37.7%), 135 (21.6%), 105 (16.8%) of the study participants got information about STI from school, health institutions, media, and peer (neighbor) respectively. Nearly two-third (61.4%) of study subjects had awareness about the transmission of STI and while 473 (75.8%) did not know the symptom of the disease. Nearly one-third of 192 (30.8%) study participants knew the common types of STI. More than half of 321 (51.4%), study participants had awareness about the prevention of mother to child transmission. From the total respondent's, nearly three-fourth (72.1%) of the participants knew that STDs could be prevented through the use of a condom. Among 624 study participants, two hundred seventeen (34.8%) of study participants had good knowledge about STI.

### Seroprevalence of syphilis

Among the total study participants, the seroprevalence of syphilis using the RPR or VDRL test was 30 (4.8%) (95% CI, [3.12, 6.48%]). The magnitude of syphilis was high in women older

**Table 1. Socio-demographic characteristics of study participants in Jinka town public health facilities, southern Ethiopia, August 21-September 9, 2020 (n = 624).**

| Variables | Category | Frequency | Percent |
|---|---|---|---|
| Age category | 18–24 | 244 | 39.1 |
| | 25–29 | 252 | 40.4 |
| | 30–42 | 128 | 20.5 |
| Marital status | Married | 588 | 94.2 |
| | Single | 16 | 2.6 |
| | Divorced | 14 | 2.2 |
| | Widowed | 6 | 1.0 |
| Religion | Orthodox | 386 | 61.9 |
| | Protestant | 204 | 32.7 |
| | Muslim | 34 | 5.4 |
| Residence | Urban | 495 | 79.3 |
| | Rural | 129 | 20.7 |
| Family type | Monogamous | 578 | 92.6 |
| | Polygamous | 46 | 7.4 |
| Education | No formal education | 123 | 19.7 |
| | Primary education | 195 | 31.3 |
| | Secondary and above | 306 | 49 |
| Partner education | No formal education | 95 | 15.2 |
| | Primary education | 199 | 31.9 |
| | Secondary and above | 330 | 52.9 |
| Occupation | Housewife | 255 | 40.9 |
| | Merchant | 156 | 25 |
| | Government employee | 153 | 24.5 |
| | Others* | 60 | 9.6 |
| Partner occupation | Merchant | 182 | 29.2 |
| | Government employee | 168 | 26.9 |
| | Farmer | 128 | 20.5 |
| | private employee | 125 | 20.0 |
| | Others | 21 | 3.4 |
| Monthly income (ETB) | 500–1000 | 60 | 9.6 |
| | 1001–2000 | 131 | 21.0 |
| | 2001–3000 | 208 | 33.3 |
| | 3001–10000 | 225 | 36.1 |

than 30 years (12.5%), pregnant women with high syphilis seroprevalence were found among rural dwelling (10.9%), who were illiterate (9.8), whose husbands were illiterate (9.5%), who were occupationally housewives (6.7%). Syphilis sero-positivity was also high in multigravida women (6.2%). Likewise, a high seroprevalence of syphilis was found on those who had two or more sexual partners (8.8%) and who drank alcohol (6.7%).

**Table 2. Medical-related characteristics of pregnant women attending ANC in Jinka town public health facilities, Southern Ethiopia, 2020.**

| Variables | Category | Frequency | Percent |
|---|---|---|---|
| History of STI | Yes | 24 | 2.9 |
| | No | 600 | 97.1 |
| HIV serostatus | Reactive | 61 | 9.8 |
| | Non-reactive | 563 | 90.2 |

**Table 3. Behavior related characteristics of study participants in Jinka town public health facilities, Southern Ethiopia, 2020.**

| Variables | Category | Frequency | Percent |
|---|---|---|---|
| Multiple sexual partners (lifetime) | Yes | 273 | 43.8 |
| | No | 351 | 56.2 |
| Multiple sexual partners (<12 months) | Yes | 49 | 7.9 |
| | No | 575 | 92.1 |
| Alcohol use | Yes | 329 | 52.7 |
| | No | 295 | 47.3 |

## Factors associated with syphilis serostatus

From bivariable logistic regression, age, residence, education, partner education, occupation, gravidity, multiple sexual partners, and use of alcohol were candidate variables for multivariable analysis having a p-value ≤0.2. To control the effects of confounder, multivariable analysis was carried out. In multiple logistic regression, rural residence, alcohol users, having multiple sexual partners were significantly associated with higher odds of seropositive for syphilis.

Women who come from rural parts of the study area were nearly three times more likely to be seropositive for syphilis than women who come from the urban area, AOR (95% CI = 2.873 (1.171, 7.050)). Pregnant women who are alcohol users had significantly greater odds of being seropositive for syphilis than women who are not alcohol users, AOR (95% CI = 3.340 (1.354, 8.241)). The odds of syphilis increased in pregnant women who had multiple sexual partners by approximately five times than women who had a single sexual partner, AOR (95% CI = 5.012 (1.929, 13.020)) (**Table 4**).

## Discussion

The overall seroprevalence of syphilis among pregnant women attending antenatal care was 4.8% with 95% CI, (3.12, 6.48%). In this study, a statistically significant association was observed between syphilis and residence, alcohol use, and having multiple sexual partners. Study participants who were from rural residences, who had multiple sexual partners and drank alcohol were the factors associated with seroprevalence of syphilis.

The finding of this study was found to be comparable with studies conducted at Yirgalem general hospital, in the Democratic Republic of Congo and Brazil which were (5.1%), (4.2%) and (4.4%) respectively [15, 18, 19].

However, the finding of this study was higher than the findings in Bahir Dar-Ethiopia, Gondar-Ethiopia, Nigeria, Bangladesh, and Hungary which were (2.6%), (2.9%), (1.98%), (2.96%), and (2.9%) consequently [12, 20–23]. The possible reason for the difference in the seroprevalence of syphilis between studies might be explained by the difference in socio-demographic characteristics, the cultural difference across the population, the data source (primary or secondary data), and methods used for diagnosis.

On the other hand, the finding of this study was lower than studies done in South Sudan and Zambia (22.1%) and (9.3%) respectively [24, 25]. The possible reason might be due to a difference in the study period and difference in population.

Pregnant women who were from the rural area were nearly three times more likely to syphilis seropositive as compared to their counterparts. The finding of this study is supported by other studies conducted in the Democratic Republic of Congo [18], Tanzania [7], Bahir Dar [20], and Gondar [26]. This might be explained by the fact that women who live in rural areas have no better access to information; education, communication, and health facilities, poor

**Table 4. Factors associated with seroprevalence of syphilis among ANC attendees in Jinka town public health facilities, southern Ethiopia, bivariable and multivariable analysis, 2020.**

| Variable | Category | Syphilis (n = 624) | | COR(95%CI) | AOR(95%CI) |
|---|---|---|---|---|---|
| | | R | NR | | |
| **Age** | <25 | 8 | 236 | 1 | 1 |
| | 25–29 | 6 | 246 | 0.720 (0.246, 2.105) | 0.419 (0.125, 1.411) |
| | 30–42 | 16 | 112 | 4.214 (1.752, 10.140) | 1.579 (0.464, 5.375) |
| **Residence** | Urban | 16 | 479 | 1 | 1 |
| | Rural | 14 | 115 | 3.645 (1.729, 7.682) | **2.873 (1.171, 7.050)** |
| **Education** | No formal education | 12 | 111 | 3.568 (1.463, 8.699) | 1.211 (0.308, 4.757) |
| | Primary education | 9 | 186 | 1.597 (0.623, 4.096) | 1.017 (0.317, 3.263) |
| | Secondary and above | 9 | 297 | 1 | 1 |
| **Partner education** | No formal education | 9 | 86 | 4.212(1.578,11.242) | 1.322 (0.354, 4.938) |
| | Primary education | 13 | 186 | 2.813 (1.145, 6.913) | 1.962 (0.709, 5.430) |
| | Secondary and above | 8 | 322 | 1 | 1 |
| **Occupation** | | | | | |
| | Housewife | 17 | 238 | 3.571 (1.029,12.394) | 1.279 (0.253, 6.460) |
| | Merchant | 7 | 149 | 2.349 (.596, 9.257) | 1.181 (0.246, 5.660) |
| | Others | 3 | 57 | 2.632 (.516, 13.419) | 1.361 (0.209, 8.864) |
| | Employee | 3 | 150 | 1 | 1 |
| **Gravida** | Prim gravida | 4 | 200 | 1 | 1 |
| | Multigravida | 26 | 394 | 3.299 (1.136, 9.584) | 2.210 (0.605, 8.079) |
| **Alcohol use** | Yes | 22 | 307 | 2.571 (1.127, 5.867) | **3.340 (1.354,8.241)** |
| | No | 8 | 287 | 1 | 1 |
| **Multiple sexual partners** | Yes | 24 | 249 | 5.542 (2.232, 13.759) | **5.012 (1.929,13.02)** |
| | No | 6 | 345 | 1 | 1 |

understanding of health service at ANC, than their urban counterparts. Moreover, cultural practices are more prominent in rural areas than in urban.

The finding in this study is contrary to a study conducted in Nigeria [21]. The possible reason for the difference in the findings might be due to the difference in population and more sexual practices common in urban than rural areas.

In this study, the risk of syphilis in pregnant women who had a history of multiple sexual partners was five times more likely to be infected than their counterparts. This is in agreement with the reports of China [27], Zambia [24], and Bahir Dar, Ethiopia [20], who noted increased odds of syphilis among women with many sexual partners as compared to those with one sexual partner. The possible reason might be explained as having many sexual partners increase vulnerability for STIs due to unsafe sexual practices and due to the low level of awareness on the transmission and prevention methods of STI.

The result of this study revealed also the presence of a significant association between alcohol use and seroprevalence of syphilis. Pregnant women who had a history of alcohol consumption were found to be about three times more likely to be seropositive for syphilis infection than those counterparts. This finding is in line with other studies conducted in Tanzania where pregnant women with a history of alcohol intake were around six times more likely to be seropositive for syphilis [28]. This might be explained by the fact that alcohol intake could be one of the numerous health-related determinant factors such women commence, including concomitant multiple sexual partnerships and a high rate of unsafe sexual practice that predisposes to the acquisition of Sexually transmitted infections (STIs) including syphilis.

## Conclusion

This study revealed that syphilis was prevalent among pregnant women, indicating that it is a significant public health problem. Study participants who were from a rural area, having multiple sexual partners and alcohol use were found to be a factor that significantly increases the risk of being infected with syphilis among pregnant women.

## Limitation of the study

Since there was card review to address some variables there may have missing data and by the fact that they are carried out at one time point and give no indication of the sequence of events so, it is impossible to infer causality.

## Supporting information

**S1 File. Data collection tool.**
(DOCX)

**S2 File. The dataset used for this study.**
(SAV)

## Author Contributions

**Conceptualization:** Mulusew Enbiale, Frehiwot Haile, Direslgn Misekir.

**Data curation:** Mulusew Enbiale, Direslgn Misekir.

**Formal analysis:** Mulusew Enbiale, Asmare Getie, Frehiwot Haile, Direslgn Misekir.

**Funding acquisition:** Mulusew Enbiale.

**Investigation:** Mulusew Enbiale, Asmare Getie, Frehiwot Haile, Beemnet Tekabe, Direslgn Misekir.

**Methodology:** Mulusew Enbiale, Asmare Getie, Frehiwot Haile, Beemnet Tekabe, Direslgn Misekir.

**Project administration:** Mulusew Enbiale, Direslgn Misekir.

**Resources:** Mulusew Enbiale, Frehiwot Haile, Direslgn Misekir.

**Software:** Mulusew Enbiale, Asmare Getie, Frehiwot Haile, Beemnet Tekabe, Direslgn Misekir.

**Supervision:** Mulusew Enbiale, Asmare Getie, Frehiwot Haile, Beemnet Tekabe, Direslgn Misekir.

**Validation:** Mulusew Enbiale, Asmare Getie, Frehiwot Haile, Direslgn Misekir.

**Visualization:** Mulusew Enbiale, Asmare Getie, Frehiwot Haile, Beemnet Tekabe, Direslgn Misekir.

**Writing – original draft:** Mulusew Enbiale, Asmare Getie, Frehiwot Haile, Beemnet Tekabe, Direslgn Misekir.

**Writing – review & editing:** Mulusew Enbiale, Asmare Getie, Frehiwot Haile, Direslgn Misekir.

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
