## [Decision Letter · Decision Letter 0]

30 Jul 2021

PONE-D-21-19327

Magnitude of Syphilis Sero-status And Associated Factors among Pregnant Women Attending Antenatal Care in Jinka Town Public Health Facilities, Southern Ethiopia, 2020

PLOS ONE

Dear Dr. Getie,

Thank you for submitting your manuscript to PLOS ONE. After careful consideration, we feel that it has merit but does not fully meet PLOS ONE’s publication criteria as it currently stands. Therefore, we invite you to submit a revised version of the manuscript that addresses the points raised during the review process.

We look forward to receiving your revised manuscript.

Kind regards,

Jianguo Wang, PhD

Academic Editor

PLOS ONE

2. lease include additional information regarding the survey or questionnaire used in the study and ensure that you have provided sufficient details that others could replicate the analyses. For instance, if you developed a questionnaire as part of this study and it is not under a copyright more restrictive than CC-BY, please include a copy, in both the original language and English, as Supporting Information. If the original language is written in non-Latin characters, for example Amharic, Chinese, or Korean, please use a file format that ensures these characters are visible.

3. Please state whether you validated the questionnaire prior to testing on study participants. Please provide details regarding the validation group within the methods section.

4. Please amend your current ethics statement to address the following concerns: Please explain why written consent was not obtained, how you recorded/documented participant consent, and if the ethics committees/IRBs approved this consent procedure.

5. Please note that in order to use the direct billing option the corresponding author must be affiliated with the chosen institute. Please either amend your manuscript to change the affiliation or corresponding author, or email us at plosone@plos.org with a request to remove this option

7. We note you have included a table to which you do not refer in the text of your manuscript. Please ensure that you refer to Table 2 in your text; if accepted, production will need this reference to link the reader to the Table.

8. Please include a copy of Table 4 which you refer to in your text on page 11.

Reviewers' comments:

Reviewer's Responses to Questions

**Comments to the Author**

1. Is the manuscript technically sound, and do the data support the conclusions?

Reviewer #1: Yes

2. Has the statistical analysis been performed appropriately and rigorously? 

Reviewer #1: Yes

3. Have the authors made all data underlying the findings in their manuscript fully available?

Reviewer #1: Yes

4. Is the manuscript presented in an intelligible fashion and written in standard English?

Reviewer #1: Yes

5. Review Comments to the Author

Reviewer #1: REVIEW

of manuscript number PONE-D-21-19327, entitled “ Magnitude of Syphilis Sero-status And Associated Factors among Pregnant Women Attending Antenatal Care in Jinka Town Public Health Facilities, Southern Ethiopia, 2020”

The manuscript submitted for review provides epidemiological information on the distribution of syphilis among pregnant women in an Ethiopian town. As well stated in the Introduction Section, the disease is one of the most important global health problems. As the Sub-Saharan population is of increased risk with relatively high seroprevalence, information about syphilis epidemiology is urgently needed. The work fits well in the scope of the journal and although this is a simple analysis that adds nothing to the global current knowledge, it may provide useful information for the national and regional health authorities to implement more adequate prevention and information programs. The paper is written in simple and very clear language that makes it readable by a broad target audience (however language and style editing is needed; someone should carefully read the manuscript and correct the grammar errors). Additionally, all analyses used are well and sufficiently described.

However, the following deficiencies should be indicated:

• The study design and the paper design are very similar to those of an already published work dealing with the same research problem (High seroprevalence of syphilis infection among pregnant women in Yiregalem hospital southern Ethiopia by Anteneh Amsalu, Getachew Ferede and Demissie Assegu, published in BMC Infectious Diseases, 2018, 18). I have the impression that the authors have taken the above-mentioned paper and have applied the same design to a different population.

• The confirmatory character of the work. The paper confirms well-known facts and no novel associations of syphilis seroprevalence in pregnant were found.

• The limitations of the work are not stated and discussed in the Discussion section.

• Abbreviations, such as ANC, CS, COR, AOR, although clear, should be clarified at the first time of use.

• Figures 2 & 3 are unnecessary. The information given in the text is enough.

6. PLOS authors have the option to publish the peer review history of their article (what does this mean?). If published, this will include your full peer review and any attached files.

Reviewer #1: **Yes: **Tatina Todorova

---

## [Author Response · Author response to Decision Letter 0]

20 Aug 2021

Author’s Point-by-Point Response to the Reviewer's and Editors Reports

Title: Magnitude of Syphilis Sero-status And Associated Factors among Pregnant Women Attending Antenatal Care in Jinka Town Public Health Facilities, Southern Ethiopia, 2020

Corresponding author: Asmare Getie/ asmaregetie2017@gmail.com

ID: - PONE-D-21-19327

 Journal: PLOS ONE 

Point by point response to Reviewers and Editors 

First of all, the authors would like to thank PLOSE ONE Journal editors and the respective reviewers for reviewing our manuscript and providing the necessary comments to be corrected. As per the comments given, we have made corrections point by point to comment. The authors tried to answer all the issues raised by editorial team and reviewers. Please note that we gave our response in blue font color.

Comment 1: please include additional information regarding the survey or questionnaire used in the study and ensure that you have provided sufficient details that others could replicate the analyses.

Response 1: the questionaries’ were putted as supporting information in the manuscript and it was submitted to the journal as supporting information 

Comment 2: Please state whether you validated the questionnaire prior to testing on study participants. Please provide details regarding the validation group within the methods section

Response 2: The questionnaire was already validated hand used by other previous studies 

Comment 3. Please amend your current ethics statement to address the following concerns: Please explain why written consent was not obtained, how you recorded/documented participant consent, and if the ethics committees/IRBs approved this consent procedure.

Response 3: Thank you for this suggestion, actually it was editorial error, we have taken written consent from the study participants. It was corrected in the manuscript part.

Comment 4. Please note that in order to use the direct billing option the corresponding author must be affiliated with the chosen institute. Please either amend your manuscript to change the affiliation or corresponding author, 

Response 4: thank you very much, it was corrected accordingly. The corrected one was appreciated in the manuscript.

Comment 5: In your Data Availability statement, you have not specified where the minimal data set underlying the results described in your manuscript can be found. PLOS defines a study's minimal data set as the underlying data used to reach the conclusions drawn in the manuscript and any additional data required to replicate the reported study findings in their entirety. All PLOS journals require that the minimal data set be made fully available. For more information about our data policy, please see http://journals.plos.org/plosone/s/data-availability.

Response 5: thank you very much for this suggestion, the data was mentioned in the manuscript under supporting information and it was uploaded to the journal as supporting information.

Comment 6. We note you have included a table to which you do not refer in the text of your manuscript. Please ensure that you refer to Table 2 in your text; if accepted, production will need this reference to link the reader to the Table.

Response 6: thank you, it is corrected accordingly 

Comment 7. Please include a copy of Table 4 which you refer to in your text on page 11

Response 7: thank you, it was corrected as accordingly 

Comment 8: The limitations of the work are not stated and discussed in the Discussion section.

Response 8: thank you very much, as per your recommendation the limitations of this study was incorporated in the manuscript part 

comment 9: Abbreviations, such as ANC, CS, COR, AOR, although clear, should be clarified at the first time of use.

Response 9: those abbreviations were clarified accordingly 

comment 10: Figures 2 & 3 are unnecessary. The information given in the text is enough.

Response 10: thank you very much as per your recommendation figures 2 and 3 were excluded

---

## [Decision Letter · Decision Letter 1]

31 Aug 2021

Magnitude of Syphilis Sero-status And Associated Factors among Pregnant Women Attending Antenatal Care in Jinka Town Public Health Facilities, Southern Ethiopia, 2020

PONE-D-21-19327R1

Dear Dr. Getie,

We’re pleased to inform you that your manuscript has been judged scientifically suitable for publication and will be formally accepted for publication once it meets all outstanding technical requirements.

Kind regards,

Jianguo Wang, PhD

Academic Editor

PLOS ONE

Additional Editor Comments (optional):

Reviewers' comments:

Reviewer's Responses to Questions

**Comments to the Author**

1. If the authors have adequately addressed your comments raised in a previous round of review and you feel that this manuscript is now acceptable for publication, you may indicate that here to bypass the “Comments to the Author” section, enter your conflict of interest statement in the “Confidential to Editor” section, and submit your "Accept" recommendation.

Reviewer #1: All comments have been addressed

2. Is the manuscript technically sound, and do the data support the conclusions?

Reviewer #1: Yes

3. Has the statistical analysis been performed appropriately and rigorously? 

Reviewer #1: Yes

4. Have the authors made all data underlying the findings in their manuscript fully available?

Reviewer #1: Yes

5. Is the manuscript presented in an intelligible fashion and written in standard English?

Reviewer #1: Yes

6. Review Comments to the Author

Reviewer #1: The authors have adequately addressed all comments raised in a previous round of review and I don't have any additional comments.

7. PLOS authors have the option to publish the peer review history of their article (what does this mean?). If published, this will include your full peer review and any attached files.

Reviewer #1: **Yes: **Tatina Todorova

---

## [Editor Report · Acceptance letter]

2 Sep 2021

PONE-D-21-19327R1 

Magnitude of Syphilis Sero-status And Associated Factors among Pregnant Women Attending Antenatal Care in Jinka Town Public Health Facilities, Southern Ethiopia, 2020 

Dear Dr. Getie:

I'm pleased to inform you that your manuscript has been deemed suitable for publication in PLOS ONE. Congratulations! Your manuscript is now with our production department. 

Kind regards, 

on behalf of

Dr. Jianguo Wang 

Academic Editor

PLOS ONE